# Molecularly Imprinted Polymer-Amyloid Fibril-Based Electrochemical Biosensor for Ultrasensitive Detection of Tryptophan

**DOI:** 10.3390/bios12050291

**Published:** 2022-05-02

**Authors:** Ibrar Alam, Benchaporn Lertanantawong, Thana Sutthibutpong, Primana Punnakitikashem, Piyapong Asanithi

**Affiliations:** 1Nanoscience and Nanotechnology, Faculty of Science, King Mongkut’s University of Technology Thonburi, Bangkok 10140, Thailand; ibrar.alam@mail.kmutt.ac.th; 2Department of Biomedical Engineering, Faculty of Engineering, Mahidol University, Salaya 73170, Thailand; benchaporn.ler@mahidol.ac.th; 3Department of Physics, Faculty of Science, King Mongkut’s University of Technology Thonburi, Bangkok 10140, Thailand; thana.sut@mail.kmutt.ac.th; 4ThEP Center, Commission of Higher Education, 328 Si Ayuthaya Rdad, Bangkok 10400, Thailand; 5Theoretical and Computational Science Center (TaCS), Science Laboratory Building, Faculty of Science, King Mongkut’s University of Technology Thonburi, Bangkok 10140, Thailand; 6Siriraj Center of Research Excellence in Theranostic Nanomedicine, Faculty of Medicine, Siriraj Hospital, Mahidol University, Bangkok 10700, Thailand; primana.pun@mahidol.ac.th; 7Department of Biochemistry, Faculty of Medicine, Siriraj Hospital, Mahidol University, Bangkok 10700, Thailand

**Keywords:** electrochemical impedance spectroscopy, molecularly imprinted polymer, amyloid fibril, sensor, tryptophan

## Abstract

A tryptophan (Trp) sensor was investigated based on electrochemical impedance spectroscopy (EIS) of a molecularly imprinted polymer on a lysozyme amyloid fibril (MIP-AF). The MIP-AF was composed of aniline as a monomer chemically polymerized in the presence of a Trp template molecule onto the AF surface. After extracting the template molecule, the MIP-AF had cavities with a high affinity for the Trp molecules. The obtained MIP-AF demonstrated rapid Trp adsorption and substantial binding capacity (50 µM mg^−1^). Trp determination was studied using non-Faradaic EIS by drop drying the MIP-AF on the working electrode of a screen-printed electrode. The MIP-AF provided a large linear range (10 pM–80 µM), a low detection limit (8 pM), and high selectivity for Trp determination. Furthermore, the proposed method also indicates that the MIP-AF can be used to determine Trp in real samples such as milk and cancer cell media.

## 1. Introduction

Tryptophan (Trp) is an essential amino acid that acts as a precursor for niacin, melatonin, and serotonin. Determining Trp can help us monitor the creation of a toxic substance in the brain, which can cause schizophrenia, delusions, and hallucinations [1,2]. Furthermore, the pattern of Trp usage observed in cancer growth media can be utilized as a signal for monitoring metastasis (aggressiveness) [3]. As a matter of fact, devising a simple approach to make Trp determination easier is important. Several techniques for determining Trp have been developed, including liquid chromatography with fluorescence detection [4], chemiluminescence [5], gas chromatography–mass spectrometry [6], high-performance liquid chromatography [7], and infrared spectroscopy [8]. These techniques can determine trace quantities of Trp because of their great sensitivity. The electrochemical technique is an alternative technique that has been extensively employed because of its high sensitivity, fast response time, miniaturization capability, low cost, low power consumption, and significant detection limits. However, direct determination of Trp from its electrochemical oxidation may struggle with low sensitivity and selectivity.

Recent research has demonstrated that molecularly imprinted polymers (MIPs) have the potential to be the most significant recognition probes in the development of electrochemical sensors. They employ a “lock-and-key” mechanism to selectively bind the molecule with which they are devised [9,10]. However, using traditional techniques, it is challenging to extract original template molecules that have become tucked away in the inside of bulk materials [11,12]. Consequently, it leads to some flaws, such as poor imprinting capacity, low adsorption rate, high nonspecific adsorption, and low physiochemical stability. To tackle these concerns, a nanoscale MIP can be devised by incorporating a nanomaterial as a supporting material during the synthesis of the imprinted material [11]. As nanomaterials may significantly enhance specific surface area and physicochemical characteristics, a nanostructured MIP with a high surface area may allow for more comprehensive template extraction, easier site access, decreased mass transfer resistance, and well-defined material topology [12,13,14]. Gold nanoparticles [15], quantum dots [16], carbon nanotubes [17], graphene [18], magnetic nanoparticles [19], and rare-earth nanoparticles [20] are examples of commonly utilized nanomaterials.

In this research, protein amyloid fibril (AF) was investigated in an attempt to produce a new supporting material. Compared to other types of supporting materials, AF may be a good choice as the core material of an MIP due to its unique properties such as tensile strength [21] and high surface area for molecular adsorption [11]. In addition, the plethora of functional residues on the AF surface, such as amines, carboxylic acids, and sulfhydryl, enables chemical and physical interactions to broaden their functional profile [22,23,24,25,26]. AF was also utilized as an organic matrix for the fabrication of graphene [26,27], silica nanoparticles [28], metal nanostructures [29], and calcium phosphate minerals [30], as well as titania, leading to bioinorganic nanocomposites with better sensing and energy conversion characteristics [31]. Using electrostatic interactions, AF was decorated with sulfonated polythiophene to create electrically and electrochemically active nanowire networks [22,32]. AF can even be functionalized chemically before or after construction, resulting in novel nanomaterials with synergistic characteristics, such as spider silk conjugated with DNA to form hierarchical structures [33], whey protein nanofibrils/glucose oxidase for glucose sensors [34]. and peptides modified with aniline residues for detecting the hepatitis B viral gene [35].

A monomer with excellent biocompatibility and multifunctional groups is also required for use as a polymer matrix in the molecular imprinting process, in addition to adequate supporting nanomaterials [11]. Polyaniline (PAni) is of great interest because of its potential to produce a wide range of supramolecular structures with different characteristics depending on the synthesis circumstances (pH, potential, molar ratio, etc.) [36,37,38]. For example, PAni polymerized at different pH levels provides different electrical conductivity values, ranging from 10^−10^ to 10^2^ S/cm [39,40]. Aniline polymerized in neutral and slightly acidic conditions has a unique supramolecular configuration and is particularly interesting to produce protein MIPs [41]. Additionally, its multifunctional groups (catechol and amino groups), biocompatibility, and hydrophilicity make it suitable for imprinting amino acids [37,38].

In this study, lysozyme amyloid fibril (AF) was fabricated with polyaniline (PAni) in the presence of Trp as a template. The embedded Trp was washed out later, leaving the cavities behind leading to the formation of a molecularly imprinted polymer-amyloid fibril (MIP-AF). The AF formation and polymerization of aniline surrounding it were confirmed using atomic force microscopy and UV–VIS spectroscopy. The electrochemical behavior of the MIP-AF was investigated by non-Faradaic EIS. The MIP-AF was used to determine Trp in phosphate-buffered saline, a milk sample, and cancer cell culture media.

## 2. Materials and Methods

### 2.1. Protein Amyloid Fibril (AF) Formation

First, 2 mM lysozyme (Lyz) from hen egg white (Sigma-Aldrich Inc., St. Louis, MA, USA) solution was prepared in DI water, and by adding HCl (Sigma-Aldrich, Inc.), the pH was adjusted to 2. The solution was incubated for 3 days at 65 °C. The development of a hydrogel indicates the accomplishment of AF production, which was confirmed by AFM. No particles corresponding to unconverted protein were found in the AFM results, suggesting that protein was nearly completely converted to nanofibers [42].

### 2.2. Aniline Polymerization

#### Preparation of MIP-AF

The specified quantity (20 mM) of aniline was added to the 2 mM AF. To begin the oxidative polymerization, 10 mM FeCl_3_ in 10 mM HCl was added and properly mixed, yielding a desired final nanofiber concentration of 0.25 mM. After initiating the polymerization of aniline around AF, the Trp (template molecule) was introduced at different time intervals. The polymerization time was used to regulate the aniline polymer thickness around the AF. For the next 5 h, the polymerization phase was carried out at room temperature. The product was rinsed with DI water after polymerization to eliminate the free monomer, rinsed with a solution of ethanol (1:4, *v*/*v*) and hydrochloric acid (1 mol L^−1^) to remove the Trp template, and then thoroughly washed with DI water. The imprinted polymerized amyloid fibril was designated as MIP-AF. The nonimprinted AF (NIP-AF, a control sample) was made in the same manner as the molecularly imprinted polymer-AF (MIP-AF), but without the Trp template molecule.

### 2.3. Methods

#### 2.3.1. UV–VIS Spectroscopy

A UV–VIS spectrophotometer (Aquamate Thermo Fisher Scientific, Beijing, China) was used to record the absorbance spectra with a path length of 1 cm.

#### 2.3.2. AFM Microscopy

AFM images were obtained employing SiN cantilevers with noncontact mode and a spring constant of 2 N/m on a Park System (AFM, Park NX10 instrument). The obtained images were processed and analyzed with XEI ver. 1. 7. 6.

#### 2.3.3. Electrochemical Impedance Spectroscopy

Electrochemical impedance spectroscopy (EIS) was performed by employing the VersaSTAT4 potentiostat of Princeton Applied Research. Furthermore, Dropsense-D. S. C. was employed as a boxed connection for a screen-printed electrode (SPE., by CI1703OR-Quasense). The counter electrode (CE) and working electrode (WE, 3 mm in diameter) in an SPE are made of carbon paste, whereas the reference electrode (RE) is made of silver/silver chloride (Ag/AgCl). The EIS results were obtained at room temperature [43].

The blank WE was regarded as a nonmodified SPE. A 4 μL volume of the AF, aniline monomers, PAni, and NIP-AF and MIP-AF solutions was dropped on the working electrode of the SPE for EIS analysis. The following input metrics were used in EIS assays: At open-circuit potential, the frequency range was 100 kHz–0.1 Hz, with an amplitude of 10 mV. The experiment was performed five times under each condition, with the average and standard deviation determined. Tyrosine (Tyr) and phenylalanine (Phe) were used as comparative amino acids with initial concentrations of 5 mM to investigate the MIP-AF selectivity. A Randles equivalent circuit was used to simulate the impedance data using Versa-Studio 2.60.6 software. EIS analysis was performed using equivalent circuits in order to model the impedance properties through the MIP-AF interface.

#### 2.3.4. Adsorption Study by Binding Analysis

In a 1.5 mL centrifuge tube, 1.0 mg of the MIP-AF or NIP-AF was mixed with various concentrations of Trp in phosphate-buffered saline (PBS, pH 7.4). The samples were centrifuged for 5 min at 10,000 rpm after 1 h of stirring at room temperature. The Trp concentration in the supernatant was determined by UV–VIS absorption spectroscopy, which had a distinctive peak of 405 nm. The amount of Trp adsorption (Q) by the MIP-AF or NIP-AF was determined by using the following equation:Q = (C_0_ − C) V/M(1)

Here, the initial Trp concentration (µM) is denoted by C_0_, the concentration of Trp upon adsorption is C, the volume of Trp solution (µL) is V, and M denotes the weight of the MIP-Trp or NIPs (mg). With an initial concentration of Trp of 5 µM, the adsorption kinetics were assessed by altering the time for adsorption from 0 to 120 min.

#### 2.3.5. Real-Sample Preparation for Trp Detection

The developed MIP-AF sensor’s basic and clinical applicability for determining Trp in cow’s milk and cancer cell culture media was tested. The MIP-AF sensor’s results were related to the analysis of high-performance liquid chromatography (HPLC). The HPLC system was Shimadzu Prominence UFLC (Pump: LC-20 AD; Detector: SPD-M20A; Column: Inertsil ODS-3 (GL Sciences Inc., San Diego, CA, USA) 5 μm in diameter, C-18, 4.6 × 250 mm). The mobile phase was acetate-buffered pH 7.0 and acetonitrile (9:1 *v*/*v*) and detection at 225 nm. Before analysis, the milk sample was dissolved 10-fold with 0.01 M PBS (pH 7.0), and McCoy cancer cell culture media was used as is.

The human colorectal cancer cell line Ht29 was cultivated on 6-well plates provided by McCoy comprising 10% FBS, penicillin G (100 U/mL), and streptomycin (100 mg/mL) at 37 °C and 5% CO_2_. The seeding density per well was 3 × 10^6^ cells. The trend of Trp metabolism was examined using the MIP-AF by analyzing supernatants and testing them immediately, as well as after 12, 24, and 48 h post-culture.

## 3. Results

### 3.1. Material Characterization

AFM height images of bare amyloid fibril (AF), nonimprinted AF (NIP-AF, a control sample without imprinted Trp), molecularly imprinted polymer-amyloid fibril without Trp extraction (TrpIP-AF), and molecularly imprinted polymer-amyloid fibril after Trp extraction (MIP-AF) are presented in Figure 1. AFs several micrometers long and with a mean diameter of 9 ± 0.9 nm were observed, as shown in Figure 1a. Figure 1b shows that the AF was coated by polymerizing aniline around it to form the NIP-AF with a mean diameter of 31 ± 1.1 nm. Here, the AF surface performs as a nucleation site for inducing PAni polymerization. Without AF, PAni polymerization forms a cluster-like structure (Appendix A). In Figure 1c, the average diameter of the TrpIP-AF (Trp template still present) is about 30 ± 4.2 nm, while in Figure 1d, the average diameter of the MIP-AF is 30 ± 2.1 nm. The TrpIP-AF and MIP-AF have the same average diameter, implying that the AFM study may not be able to distinguish the presence and absence of Trp templates in the TrpIP-AF and MIP-AF.

UV–VIS spectroscopy was employed to investigate the formation of PAni on the AF surface, as well as to confirm the presence and absence of Trp templates in the TrpIP-AF and MIP-AF, respectively. Interestingly, we found that the AF surface not only provided the site for PAni nucleation but also alternated the protonation state of PAni. In Figure 2a, the absorption peak at 280 nm of the AF (green line) refers to the aromatic amino acids found in the protein fibril such as tryptophan, phenylaniline, and tyrosine [44]. Polymerized PAni in the absence of the AF (black line) shows two main peaks at around 330 nm (electron transition in the benzenoid rings) and 670 nm (exciton absorption in the quinoid rings), indicating the emeraldine base [45,46]. However, polymerized PAni in the presence of the AF (red line) shows three peaks at around 350, 430, and 800 nm, indicating the protonated emeraldine form [47]. Here, the peak at 330 nm was shifted to 350 nm, and the peak at 670 nm disappeared. Peaks at 430 and 800 nm were formed due to the proton doping level and the polaronic transitions of PAni [47].

Figure 2b reveals the distinctive peak for the aromatic ring of Trp at 280 nm (green line). The AF also presents the absorbance peak at 280 nm (Figure 2a), which is the distinctive property of the peptides comprising aromatic amino acids. In the presence of the Trp template on the surface (TrpIP-AF), a distinctive peak was observed at 280 nm (blue line). Once Trp was washed out leaving the cavities behind, the peak at 280 nm disappeared (Figure 2b, red line), leading to the formation of the MIP-AF.

Additionally, we also determined different concentrations of aniline for polymerization around the AF, as shown in Appendix A. The low concentrations (5–50 mM) of aniline produce absorption peaks at around 430 and 800 nm, indicating the formation of a protonated emeraldine form, whereas a higher concentration (100 mM) of aniline provides a peak at 680 nm.

### 3.2. Electrochemical Determination of Trp

Figure 3 depicts the impedance spectra in the form of Nyquist plots, where the semicircle section refers to the electron transfer limiting process at higher frequencies, and the linear component relates to diffusion at lower frequencies. As a result, the semicircle’s diameter represents the charge transfer resistance (R_ct_), which is linked to the dielectric and insulating properties of the electrode/electrolyte interface [43].

Appendix A shows that the radius of the semicircle for the nonmodified SPE (black dot) decreased when it was modified with the MIP-AF (green dot), and the radius increased when PAni or the AF alone were introduced. Here, the semicircle diameter implies the R_ct_ value. Moreover, the EIS approach was used to determine Trp binding to the MIP-AF. The impedance spectra of the Trp analyte at various concentrations (1 × 10^−5^–2 × 10^−3^ µM) are shown in Figure 3a as a Nyquist plot. The diameter of the semicircle increased with increasing Trp concentration, indicating that the Trp interaction on the MIP-AF-modified electrode surface increased the Rct. Figure 3b shows the impedance spectra of the further higher concentration (4 × 10^−3^–80 µM) of Trp. The semicircle diameter continues to increase due to the Trp interaction on the MIP-AF-modified electrode surface.

Trp was determined by using the MIP-AF-modified electrode. Figure 4a shows the dependency of R_ct_ on Trp concentration (0.00001–80 μM). The concentrations were divided into two linear ranges, i.e., a lower concentration range of 0.00001–0.002 μM and a higher concentration range of 0.004–80 μM. The MIP-AF signals increased linearly with increasing Trp concentrations in the ranges of 0.00001–0.002 μM (Figure 4c) and 0.004–80 μM (Figure 4b) under optimal conditions. The linear equation for lower concentrations was R_ct_ = 175,270 X + 150.2, and the regression coefficient was 0.99 (Figure 4c). For higher concentrations, the linear equation was R_ct_ = 13.25 X + 558.3, and the coefficient for regression was 0.99. The LOD determined from the calibration curve (Figure 4c) was 8 pM (*n* = 5), which was lower than previously reported results (summarized in Table 1). The excellent results of the MIP-AF may be attributable to the unique surface properties of the AF: (1) the AF has a higher surface area; (2) in aqueous solution, the specific pockets on the surface of the AF are suitable for fabricating aniline, leading to emeraldine polymers.

### 3.3. Interference Study

In order to evaluate the selectivity and specificity of the MIP-AF biosensor, 0.5 μM of tyrosine (Tyr) and phenylalanine (Phe) was used to determine the MIP-AF biosensor in the presence or absence of 0.002 μM of Trp. Tyr and Phe were used for the interfering study because they have an aromatic structure similar to that of Trp [60]. The same experimental settings were used for EIS measurements. as discussed in Section 2.3.3. The R_ct_ increase was determined using the following equation: θ (%) = [(R_ct_ before–R_ct_ after)/ R_ct_ after] × 100, where “θ” represents the percent increase in R_ct_, “R_ct_ before” represents the R_ct_ of the MIP-AF, and “R_ct_ after” is R_ct_ after applying the interfering amino acids to the modified electrode, i.e., MIP-AF/SPE.

As stated in Figure 4d, in the presence of Trp (Trp, Trp + Tyr, Trp + Phe) and the absence of Trp (Tyr, Phe) produced a significantly different response. As a result, the data show that the sensor MIP-AF/SPE established could be used to determine Trp with high specificity and selectivity. According to a previous theoretical study [61], PAni on the AF in the protonated emeraldine form with surrounding negative counterions could strongly interact with Trp in the zwitterionic form. The specificity of Trp detection arose from interactions between the N-H groups of the sidechain indoles and aromatic rings of the oxidized PAni chains. The N-H group of a Trp sidechain possessed the strongest dipole moment when compared to the sidechain functional groups of Tyr and Phe [62]. As the polymerization of PAni on the AF produced under low pH resulted in the protonated emeraldine form, the AF could play a substantial role on the enhanced Trp specificity. Two critical aspects of the MIP can be emphasized based on the interference study: (1) the specificity of the cavity generated; (2) the conformational adjustment of the template and material used for the MIP. These two traits may synergistically provide a high degree of specificity in this study.

In Table 1, the proposed method for the detection of Trp is compared with previously published electrochemical methods. In previous studies, several types of electrodes were fabricated with different nanomaterials to make the surface specifically selective for the Trp analyte. To our knowledge, the proposed MIP-AF-modified SPE sensor has the lowest LOD, i.e., 8 × 10^−12^, compared to previous studies and a wide linear detection range, i.e., 1 × 10^−5^–2× 10^−3^ and 4 × 10^−3^–80 µM.

### 3.4. Adsorption Isotherms and Adsorption Kinetics

The binding investigations were carried out at various concentrations of Trp, ranging from 1 to 120 µM, to assess the binding ability of the MIP-AF against control NIP-AF (see Section 2.3.4). Figure 5a demonstrates the adsorption isotherm of the MIP-AF and NIP-AF, which shows that the MIP-AF had higher capability of binding than that of NIP-AF to Trp. A saturation value was achieved by the Trp concentration at 80 µM. The maximum adsorption capacity in the experiments is determined to be 50.1 µM/mg, 7.2 µM/mg for the MIP-AF and NIP-AF, respectively. The binding ratio of the MIP-AF/NIP-AF was found to be 6.95, known as the imprinting factor.

The kinetics of adsorption were investigated further, and the results are presented in Figure 5b. The MIP-AF absorbed 79% of the equilibrium quantity in about 5 min, and the overall equilibrium period was less than 1 h. The imprinted material’s rebinding rate is extremely fast, especially when compared to earlier imprinted materials that took more than 2 h to reach the adsorption plateau [11,63].

The NIP-AF contains no imprinted cavities and thus has poor resistance to nonspecific binding, as seen in Figure 5b. There are two types of imprinted cavities for the MIP-AF: surface cavities and cavities near the surface [11]. The former, which accounted for most of the adsorption due to more accessible sites and reduced mass transfer resistance, demonstrated quick equilibrium adsorption. The latter, when investigating the surface imprinting technique and the superthin coating of PAni produced on the AF, also exhibited reasonable rebinding kinetics capability.

### 3.5. Stability

A stock of MIP-AF-modified SPEs was stored at room temperature for ten weeks to assess storage stability. During this period, the signal was maintained consistently (Figure 6).

The biosensors’ storage stability was determined by the following equation:(2)$$\mathrm{Stability}\;(\%)=\frac{\left|R_{\mathrm{ct}\;\left(\mathrm{fresh}\right)}-R_{\mathrm{ct}\;\left(10\;\mathrm{weeks}\right)}\right|}{R_{\mathrm{ct}\;\left(\mathrm{fresh}\right)}}\times \;100\%$$
where R_ct(fresh)_ is obtained from the freshly prepared MIP-AF-modified SPEs, and R_ct(10 weeks)_ is obtained from the modified SPEs stored at room temperature for 10 weeks. The storage stability of the modified SPEs calculated from each concentration of Trp in Figure 6 was about 98 %, implying a 2% drop in R_ct_ over a 10-week period, which is ideal for biosensor usage.

### 3.6. Determination of Trp in Real Samples

#### 3.6.1. Trp Consumption Pattern in Cancer Cell Media of HT 29 Cell Line

Trp levels in cancer cell culture medium were determined using the MIP-AF-modified SPE. The basic rate of Trp consumption in the human tumor cell line HT29 (colorectal carcinoma) was investigated using the proposed sensor and standard addition technique. The MIP-AF was able to accurately and precisely measure the Trp concentrations in these samples. The quantity of Trp in the supernatants of the cancer cell lines cultured media incubated for 48 h is shown in Figure 7. Figure 7a depicts the Nyquist plot obtained by utilizing the MIP-AF-modified SPE to analyze the tryptophan level in HT29 cell culture medium (fresh, after 12, 24, and 48 h), whereas Figure 7b depicts the drop in R_ct_ with consumption of Trp in the media of the cells cultured during that time frame. Furthermore, Table 2 provides the rate of Trp consumption by cancer cell lines incubated for 48 h by comparing the Trp determination through MIP-AF and HPLC.

Determination of Trp in the cell media may be useful for monitoring the Trp consumption of cancer cells. Indeed, metastatic cancer cells enhance their Trp consumption by upregulating the enzymes involved in the generation of endogenous metabolites in order to elude the immune system and broaden their proliferation and migration [64,65]. The application of the MIP-AF revealed that HT29 cells, which have a stronger potential for migration and can generate micrometastases, consumed a considerable amount of Trp.

#### 3.6.2. Determination of Trp in Milk

Table 3 shows the recovery rate, as well as other analytical parameters, for the proposed biosensor employed to determine Trp in milk as a real sample. The reproducibility between the impedimetric responses provided by the MIP-AF was investigated from the response to 0 µM, 5 µM, 10 µM, and 15 µM of Trp spiked in milk, each at three MIP-AF-modified electrodes fabricated in the same way, exhibiting great reproducibility, with an RSD (%) of 0.4, 1.9, 0.9 and 1.8, respectively. The equation RSD = (S*100)/X was used to compute the relative standard deviation (RSD), where S is the standard deviation, and X is the mean of the found concentrations. The equation for determining the recovery (%) was: recovery (%) = 100 − relative error. The low relative error of less than 2% compared to those of HPLC may provide the opportunity for using the MIP-AF as an alternative material for determining Trp in the food industry.

## 4. Conclusions

In this paper, an ultrasensitive Trp sensor fabricated from a molecularly imprinted polymer-amyloid fibril was introduced. The MIP-AF was prepared by chemical polymerization of aniline onto the AF surface in the presence of the Trp template. After template removal, the MIP-AF exhibited cavities with a high affinity for the Trp molecule. The physical property of the MIP-AF was characterized by AFM. The larger diameter of the MIP-AF than that of AF indicated the successful polymerization of aniline onto the AF surface. The chemical characteristics of polymerized aniline were investigated by UV–VIS spectroscopy. The protonated emeraldine form of PAni was found in the MIP-AF. Furthermore, the disappearance of the absorbance peak at 280 nm of the MIP-AF also referred to the success of template extraction. The Trp determination was investigated utilizing non-Faradaic EIS by coating the MIP-AF on the working electrode of the SPE. A wide linear range (10 pM–80 µM), low detection limit (8 pM), and high selectivity for Trp determination were obtained. When the MIP-AF was applied for the determination of Trp in real samples, such as milk and cancer cell culture media, it exhibited good results with low relative error compared to those of HPLC. Therefore, we anticipate that the MIP-AF might be useful in biosensors, trace enrichment of particular targets, and biochips.

## Figures and Tables

**Figure 1 biosensors-12-00291-f001:**
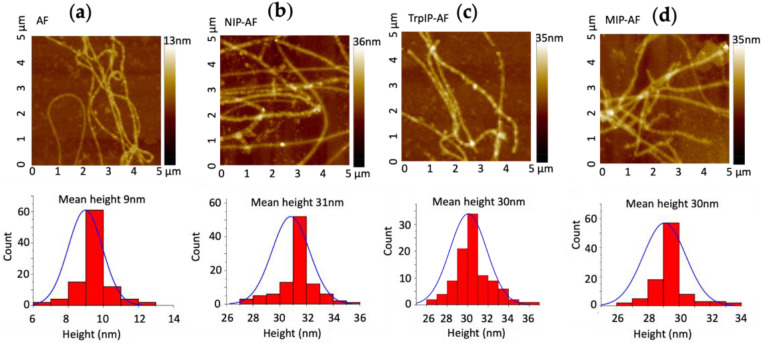
AFM topography images and height analysis; (**a**) AFM image and height analysis of amyloid fibril; (**b**) AFM image and height analysis of NIP-AF; (**c**) AFM image and height analysis of TrpIP-AF; (**d**) AFM image and height analysis of MIP-AF.

**Figure 2 biosensors-12-00291-f002:**
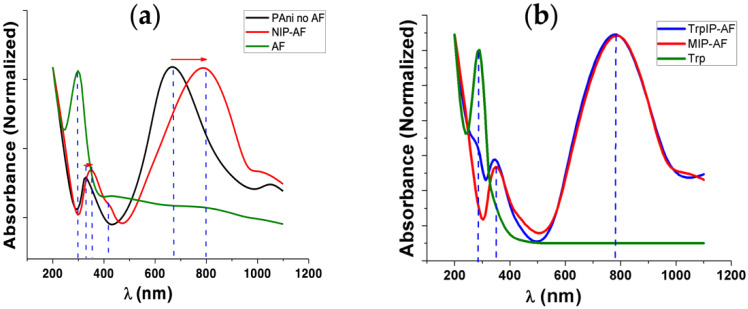
UV–VIS spectroscopy of aniline polymerization around AF; (**a**) UV–VIS spectra of bare amyloid fibril (AF, green line), aniline polymerized in the absence (Pani no AF, black line) and presence (NIP-AF, red line) of Afs; (**b**) UV–VIS of aniline polymerized around AF in the presence of template Trp on the surface (blue line), after extraction of Trp leading to MIP-AF (red line).

**Figure 3 biosensors-12-00291-f003:**
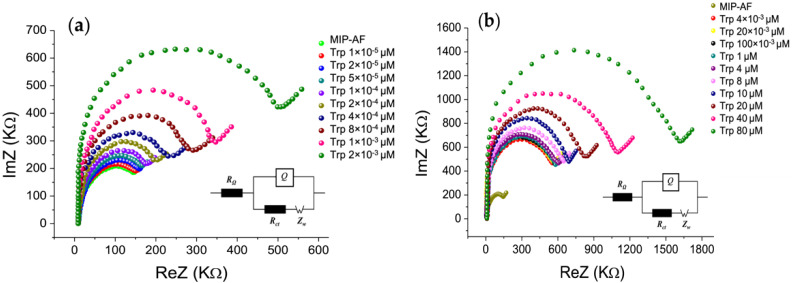
EIS Nyquist plot for characterizing MIP-AF development and its application for determining Trp in PBS; (**a**) EIS Nyquist plot for characterizing the impedance spectra of Trp analyte at lower concentrations (1 × 10^−5^–2 × 10^−3^ µM); (**b**) impedance spectra of further higher concentration (4 × 10^−3^–80 µM) of Trp. The inset shows Randle equivalent circuit, where RΩ is the solution resistance, R_ct_ is the charge transfer resistance, Q is the constant phase element, and Z_w_ is Warburg’s impedance.

**Figure 4 biosensors-12-00291-f004:**
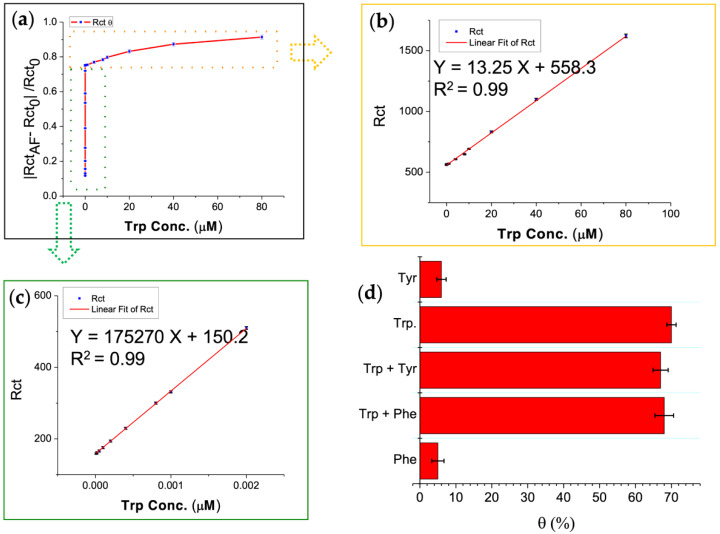
Calibration graph for analytical application and interference study carried out in 0.01 M PBS (pH 7.4); (**a**) calibration graph for determination of Trp concentration (0.00001–80 μM); (**b**) calibration graph for determination of Trp higher concentration range of 0.004–80 μM; (**c**) calibration graph for determination of Trp lower concentration range of 0.00001–0.002 μM; (**d**) specificity and selectivity of MIP-AF biosensor were determined by incubating 0.5 μM of tyrosine (Tyr) and phenylalanine (Phe) examined by MIP-AF/SPE biosensor in the presence or absence of 0.002 μM of Trp.

**Figure 5 biosensors-12-00291-f005:**
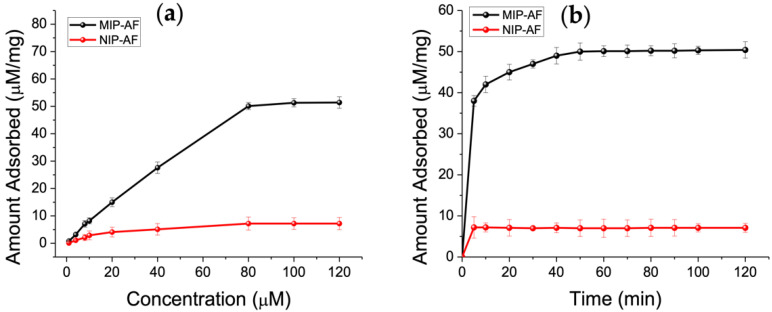
(**a**) Adsorption isotherms of Trp (1–120 µM) on MIP-AF (red) and NIP-AF(black). Amount of MIPs or NIPs: 1 mg mL1; volume: typically 1.0 mL; time: 1 h. The three measures’ means are shown by the dots. (**b**) Adsorption kinetics of Trp on MIP-AF and NIP-AF.

**Figure 6 biosensors-12-00291-f006:**
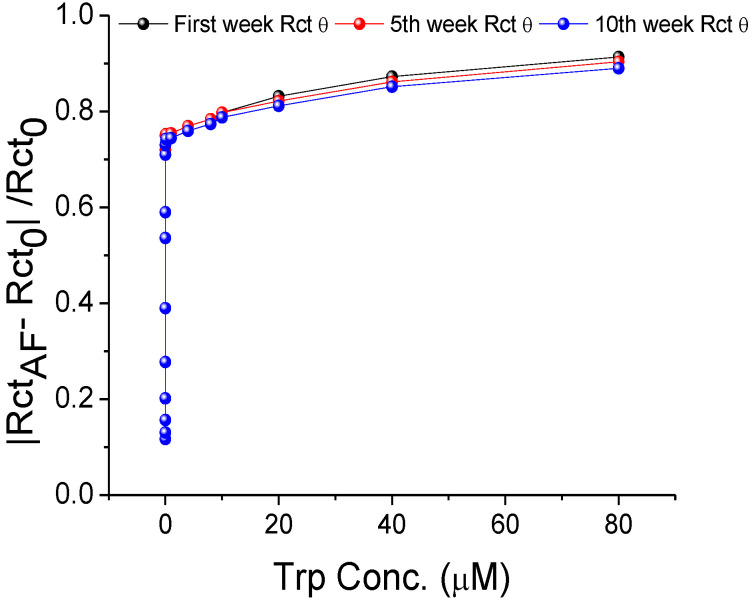
Storage stability of MIP-AF-modified SPEs at room temperature for 10 weeks.

**Figure 7 biosensors-12-00291-f007:**
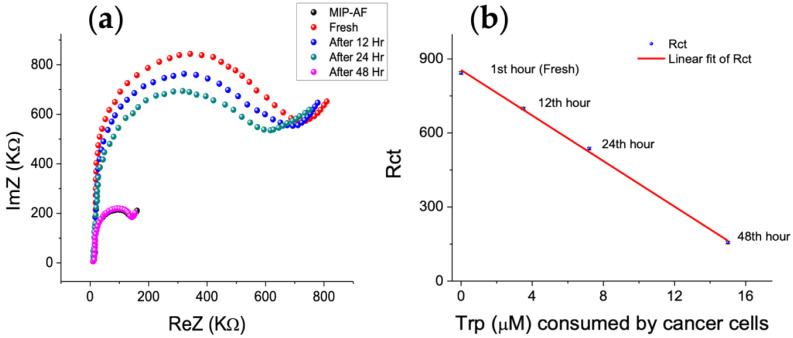
MIP-AF determination of Trp in the HT29 cell media incubated under different cellular culture periods; (**a**) Nyquist plot of Trp in PBS (black dots) and in HT29 cell culture media (fresh, after12, 24, 48 h); (**b**) decline in R_ct_ with consumption of Trp in media of the cells cultured for two days.

**Table 1 biosensors-12-00291-t001:** Comparison of the proposed method (MIP-AF/SPE) with previously published electrochemical methods used for detection of Trp.

Material and Electrode	Technique	LOD (nM)	LDR (µM)	Ref.
F-MWCNT/GCE	DPV	3.6	0.01–0.7	[48]
EGr/GCE	LSV	30.3	0.1–20	[49]
MnWO_4_/RGO/GCE	CV, DPV	4.4	0.001–120	[50]
Ta_2_O_5_-rGO-GCE	SDLSV	0.84 × 10^3^	1–8, 8–80, 80–800	[51]
PDA/RGO-MnO_2_/GCE	DPV	0.22 × 10^3^	1–300	[52]
rGO-GNPs-Cr.6/GCE	SWV	0.48× 10^3^	0.1–2.5	[53]
Apt-MWCNT/Au	CC-PSA	64 × 10^−3^	1 × 10^−4^–10, 10–300	[54]
SP-Hap-GO/SPE	LSV	5.5 × 10^3^	7–1000	[55]
MIP/ABPE	LSV	8	0.01–4, 4–20, 20–100	[56]
Nafion-MIP-MWCNTs-IL/GCE	DPV	6	8 × 10^−3^–26	[57]
MIP/GE	DPV	5	0.01–1	[58]
MIP-MWCNTs/GCE	SDLSV	1	0.002–0.2, 0.2–10, 10–100	[59]
MIP-AF/SPE	EIS	8 × 10^−3^	1 × 10^−5^–2× 10^−3^, 4 × 10^−3^–80	This work

Functionalized multiwall carbon nanotube (F-MWCNT), glassy carbon electrode (GCE), electrochemical exfoliation of graphite rods (EGr), polydopamine (PDA), reduced graphene oxide (rGO), gold nanoparticles (GNPs), hydroxyapatite graphene oxide (Hap-GO), acetylene black paste electrode (ABPE), molecularly imprinted polymer (MIP), amyloid fibril (AF), modified screen-printed electrode (SPE).

**Table 2 biosensors-12-00291-t002:** Determining Trp in human HT29 cancer cell culture media samples (*n* = 5) by HPLC and EIS with MIP-AF.

Cell Line	Incubation (h)	Cell Count	[Trp] (µM) Determined by HPLC	[Trp] (µM) Determined by MIP-AF-Modified SPE
Ht29	0	300,000 ± 87	15 ± 0.02	15 ± 0.13
	12	420,000 ± 71	10.7 ± 0.08	11.5 ± 0.05
	24	580,000 ± 93	8.1 ± 0.03	7.8 ± 0.01
	48	1,170,000 ± 52	0	0

**Table 3 biosensors-12-00291-t003:** Analyses of Trp in milk samples by the MIP-AF sensor and HPLC method (*n* = 5).

Specimen	HPLC (µM)	Spiked (µM)	Found by MIP-AF Sensor (µM)	RSD (%)	Recovery (%)	Relative Error (%)
Milk	50.23 ± 0.17	0	49.31 ± 0.22	0.4	98.2	1.8
Milk	56.11 ± 0.21	5	56.27 ± 1.07	1.9	100.3	0.3
Milk	59.28 ± 1.14	10	60.11 ± 0.53	0.9	101.4	1.4
Milk	66.13 ± 1.31	15	65.06 ± 1.21	1.8	98.4	1.6

## Data Availability

Not applicable.

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
