# Peer review of "Molecularly Imprinted Polymer-Amyloid Fibril-Based Electrochemical Biosensor for Ultrasensitive Detection of Tryptophan"

_biosensors, 2022, doi:10.3390/bios12050291_

Round 1
Reviewer 1 Report
Paper entitled “Molecular imprinted polymer-amyloid fibril based electro-chemical biosensor for ultrasensitive detection of tryptophan” meets the necessary standards for publication in this journal.
I recommend:
Attention when writing references. They are not unitary.
Please check the entire manuscript carefully for eventual typographical errors.
Final Conclusion: The paper meets the necessary standards for publication.
Reviewer 2 Report
Biosensors 1688938
The presented to Biosensors (ISSN 2079-6374) journal manuscript 1688938 is a paper about an electrochemical detection of tryptophan by biosensor.
The research plan, methods and results are well designed. However, their representation in some parts could be improved.
I would recommend major revision for this manuscript. Here are the changes that should be done:
Some general comments for whole text:
Revise the used abbreviations in order to avoid their repetition, as it is no need to explain every time when it is mentioned, i.e. AF, MIP, NIP, Trp, Lyz, etc.
Check text format for Figures captions, some captions are in bold, some are not. Another thing is that Figure caption is its title. For example (Line 112), it should not be a common phrase like “Lysozyme Amyloid Fibril (AF) was functionalized with aniline as molecular imprinted polymer (MIP)”. Change it for “A scheme of Lysozyme AF functionalized with aniline as molecular imprinted polymer (MIP)”. The same for Figure 6.
Scientific articles are written in a passive voice, it means phrases are made without pronouns. Please, remove “our” from the manuscript (line 30, 353, etc).
Unfortunately, I did not have access to your supplementary materials. Is it so large? Why not to include it the manuscript?
Other corrections:
Line 30. Not “purposed”, but proposed method.
Line 90. Should be [41]
Line 94. Scheme 1 should be removed from Introduction. It looks like a slide from a presentation. Actually, it could go to Section “Materials”, namely 2.2. You could call it Figure, not Scheme, remove graphs with results (AFM and EIS) from it, as they already shown in section “Results” and adjust its captions, explaining all three drawings.
Line 119. Change “mixing” for “adding”. Was it concentrated HCl?
Line 120 - AFM results ? no scan or picture
Line 126. Specified where or how?
Section 2.2.1 and 2.2.2 no need to separate them.
Line 139. Change section 2.3 name for “Methods”
Line 167. Define PBS.
Line 180. Specify equipment for HPLC and/or conditions for Trp detection.
Figure 1. Is it possible to improve resolution of AFM height graphs?
Line 200. Remove word “our”
Lines 324- 337. This paragraph has a lot of repetitions. Please, remove them and also a commonly known names like manganese oxide, for example. No need to explain and numerate all these abbreviations. Use the same abbreviation like rGO (instead of rGo and RGO) for the same substance/material.
Line 355. Reference needed
Line 369-371. Rewrite the paragraph. From what data did you take the value of 2%? Explain better.
Line 379. Change “Fig.7” for “Figure 7”, as it is used through the whole manuscript.
Line 396. Table 2. How many times were performed the measurements? By HPLC for example? Is it possible to indicate an interval for a measured value as in the Table 3?
Line 398. “Determination of Trp in milk”
Conclusions. You should emphasize what you did, i.e. what biosensor was produced, its characteristics and application, because this information is presented/confirmed by your measurements and results. Put the explanation “The excellent results of MIP-AF may be attributable to the unique surface properties of AF: (1) AF has a higher surface area; and (2) in aqueous solution, the specific pockets on the surface of AF are suitable for fabricating aniline leading to emeraldine state polymers.“ inside the section Results. Actually, many times it is called Results and discussion.
Good work.
Reviewer 3 Report
After reviewing this manuscript entitled: "Molecular imprinted polymer-amyloid fibril based electro-chemical biosensor for ultrasensitive detection of tryptophan", it is an interesting article.
However, I cannot recommend it to be published in Biosensors in the current form.
Some remarks have to be considered and amended before recommending acceptance for publication as follows:
1- Language has to be revised and improved. Structural mistakes arise in the manuscript, such as: (3.6.2 Determination Trp in Milk) has to be corrected to be (Determination of Trp in Milk)
2- Better resolution(s) for scheme 1 and figure 1 are recommended to show the chemical structures and legends clearly.
3- A detailed conclusion is recommended.
Round 2
Reviewer 2 Report
Thank you for making accepting changes.
Congratulation for your work.